# Inferring sparse representations of continuous signals with continuous orthogonal matching pursuit

**Karin C. Knudson**
Department of Mathematics
The University of Texas at Austin
kknudson@math.utexas.edu

**Jacob L. Yates**
Department of Neuroscience
The University of Texas at Austin
jlyates@utexas.edu

**Alexander C. Huk**
Center for Perceptual Systems
Departments of Psychology & Neuroscience
The University of Texas at Austin
huk@utexas.edu

**Jonathan W. Pillow**
Princeton Neuroscience Institute and
Department of Psychology
Princeton University
pillow@princeton.edu

## Abstract

Many signals, such as spike trains recorded in multi-channel electrophysiological recordings, may be represented as the sparse sum of translated and scaled copies of waveforms whose timing and amplitudes are of interest. From the aggregate signal, one may seek to estimate the identities, amplitudes, and translations of the waveforms that compose the signal. Here we present a fast method for recovering these identities, amplitudes, and translations. The method involves greedily selecting component waveforms and then refining estimates of their amplitudes and translations, moving iteratively between these steps in a process analogous to the well-known Orthogonal Matching Pursuit (OMP) algorithm [11]. Our approach for modeling translations borrows from Continuous Basis Pursuit (CBP) [4], which we extend in several ways: by selecting a subspace that optimally captures translated copies of the waveforms, replacing the convex optimization problem with a greedy approach, and moving to the Fourier domain to more precisely estimate time shifts. We test the resulting method, which we call Continuous Orthogonal Matching Pursuit (COMP), on simulated and neural data, where it shows gains over CBP in both speed and accuracy.

## 1 Introduction

It is often the case that an observed signal is a linear combination of some other target signals that one wishes to resolve from each other and from background noise. For example, the voltage trace from an electrode (or array of electrodes) used to measure neural activity in vivo may be recording from a population of neurons, each of which produces many instances of its own stereotyped action potential waveform. One would like to decompose an analog voltage trace into a list of the timings and amplitudes of action potentials (spikes) for each neuron.

Motivated in part by the spike-sorting problem, we consider the case where we are given a signal that is the sum of known waveforms whose timing and amplitude we seek to recover. Specifically, we suppose our signal can be modeled as:

$$y(t) = \sum_{n=1}^{N_f} \sum_{j=1}^{J} a_{n,j} f_n(t - \tau_{n,j}), \tag{1}$$

where the waveforms $f_n$ are known, and we seek to estimate positive amplitudes $a_{n,j}$ and event times $\tau_{n,j}$. Signals of this form have been studied extensively [12, 9, 4, 3].

This a difficult problem in part because of the nonlinear dependence of $y$ on $\tau$. Moreover, in most applications we do not have access to $y(t)$ for arbitrary $t$, but rather have a vector of sampled (noisy) measurements on a grid of discrete time points. One way to simplify the problem is to discretize $\tau$, considering only a finite set of possible time shift $\tau_{n,j} \in \{\Delta, 2\Delta..., N_\Delta\Delta\}$ and approximating the signal as

$$y \approx \sum_{n=1}^{N_f} \sum_{j=1}^{J} a_{n,j} f_n(t - i_{n,j}\Delta), \, i_{n,j} \in 1, ..., N_\Delta \tag{2}$$

Once discretized in this way, the problem is one of sparse recovery: we seek to represent the observed signal with a sparse linear combination of elements of a finite dictionary $\{f_{n,j}(t) := f_n(t - j\Delta), \, n \in 1, ..., N_f, \, j \in 1, ..., N_\Delta\}$. Framing the problem as sparse recovery, one can bring tools from compressed sensing to bear. However, the discretization introduces several new difficulties. First, we can only approximate the translation $\tau$ by values on a discrete grid. Secondly, choosing small $\Delta$ allows us to more closely approximate $\tau$, but demands more computation, and such finely spaced dictionary elements yield a highly coherent dictionary, while sparse recovery algorithms generally have guarantees for low-coherence dictionaries.

A previously introduced algorithm that uses techniques of sparse recovery and returns accurate and continuous valued estimates of $a$ and $\tau$ is Continuous Basis Pursuit (CBP) [4], which we describe below. CBP proceeds (roughly speaking) by augmenting the discrete dictionary $f_{n,j}(t)$ with other carefully chosen basis elements, and then solving a convex optimization problem inspired by basis pursuit denoising. We extend ideas introduced in CBP to present a new method for recovering the desired time shifts $\tau$ and amplitudes $a$ that leverage the speed and tractability of solving the discretized problem while still ultimately producing continuous valued estimates of $\tau$, and partially circumventing the problem of too much coherence.

Basis pursuit denoising and other convex optimization or $\ell_1$-minimization based methods have been effective in the realm of sparse recovery and compressed sensing. However, greedy methods have also been used with great success. Our approach begins with the augmented bases used in CBP, but adds basis vectors greedily, drawing on the well known Orthogonal Matching Pursuit algorithm [11]. In the regimes considered, our greedy approach is faster and more accurate than CBP.

Broadly speaking, our approach has three parts. First, we augment the discretized basis in one of several ways. We draw on [4] for two of these choices, but also present another choice of basis that is in some sense optimal. Second, we greedily select candidate time bins of size $\Delta$ in which we suspect an event has occurred. Finally, we move from this rough, discrete-valued estimate of timing $\tau$ to continuous-valued estimates of $\tau$ and $a$. We iterate the second and third steps, greedily adding candidate time bins and updating our estimates of $\tau$ and $a$ until a stopping criterion is reached.

The structure of the paper is as follows. In Section 2 we describe the method of Continuous Basis Pursuit (CBP), which our method builds upon. In Section 3 we develop our method, which we call Continuous Orthogonal Matching Pursuit (COMP). In Section 4 we present the performance of our method on simulated and neural data.

## 2   Continuous basis pursuit

Continuous Basis Pursuit (CBP) [4, 3, 5] is a method for recovering the time shifts and amplitudes of waveforms present in a signal of the form (1). A key element of CBP is augmenting or replacing the set $\{f_{n,j}(t)\}$ with certain additional dictionary elements that are chosen to smoothly interpolate the one dimensional manifold traced out by $f_{n,j}(t - \tau)$ as $\tau$ varies in $(-\Delta/2, \Delta/2)$.

The benefit of a dictionary that is expanded in this way is twofold. First, it increases the ability of the dictionary to represent shifted copies of the waveform $f_n(t - \tau)$ without introducing as much correlation as would be introduced by simply using a finer discretization (decreasing $\Delta$), which is an advantage because dictionaries with smaller coherence are generally better suited for sparse recovery techniques. Second, one can move from recovered coefficients in this augmented dictionary to estimates $a_{n,j}$ and *continuous*-valued estimates of $\tau_{n,j}$.

In general, there are three ingredients for CBP: basis elements, an interpolator with corresponding mapping function $\Phi$, and a convex constraint set, $C$. There are $K$ basis elements $\{g_{n,j,k}(t) = g_{n,k}(t-j\Delta)\}_{k=1}^{k=K}$, for each waveform and width-$\Delta$ time bin, which together can be used to linearly interpolate $f_{n,j}(t-\tau), |\tau| < \Delta/2$. The function $\Phi$ maps from amplitude $a$ and time shift $\tau$ to K-tuples of coefficients $\Phi(a,\tau) = (c_{n,j}^{(1)}, ..., c_{n,j}^{(K)})$, so $af_{n,j}(t-\tau) \approx \sum_{k=1}^{K} c_{n,j}^{(k)} g_{n,j,k}(t)$. The convex constraint set $C$ is for $K$-tuples of coefficients of $\{g_{n,j,k}\}_{k=1}^{k=K}$ and corresponds to the requirement that $a > 0$ and $|\tau| < \Delta/2$. If the constraint region corresponding to these requirements is not convex (e.g. in the polar basis discussed below), its convex relaxation is used.

As a concrete example, let us first consider (as discussed in [4]) the dictionary augmented with shifted copies of each waveform's derivative : $\{f_{n,j}'(t) := f_n'(t-j\Delta)\}$. Assuming $f_n$ is sufficiently smooth, we have from the Taylor expansion that for small $\tau$, $af_{n,j}(t-\tau) \approx af_{n,j}(t) - a\tau f_{n,j}'(t)$. If we recover a representation of $y$ as $c_1 f_{n,j}(t) + c_2 f_{n,j}'(t)$, then we can estimate the amplitude $a$ of the waveform present in $y$ as $c_1$, the time shift $\tau$ as $-c_2/c_1$. Hence, we estimate $y \approx c_1 f_{n,j}(t+c_2/c_1) = c_1 f_n(t - j\Delta + c_2/c_1)$. Note that the estimate of the time shift $\tau$ varies *continuously* with $c_1, c_2$. In contrast, using shifted copies of the waveforms only as a basis would not allow for a time shift estimate off of the grid $\{j\Delta\}_{j=1}^{j=N_\Delta}$.

Once a suitable dictionary is chosen, one must still recover coefficients (i.e. $c_1, c_2$ above). Motivated by the assumed sparsity of the signal (i.e. $y$ is the sum of relatively few shifted copies of waveforms, so the coefficients of most dictionary elements will be zero), CBP draws on the basis pursuit denoising, which has been effective in the compressive sensing setting and elsewhere [10],[1]. Specifically, CBP (with a Taylor basis) recovers coefficients using:

$$\text{argmin}_{\mathbf{c}} \left\| \sum_{n=1}^{N_f} (\mathbf{F_n}\mathbf{c_n^{(1)}} + \mathbf{F_n'}\mathbf{c_n^{(2)}}) - \mathbf{y} \right\|_2^2 + \lambda \sum_{n=1}^{N_f} \left\| \mathbf{c}_n^{(1)} \right\|_1 \text{ s.t. } c_{n,i}^{(1)} \geq 0 \text{ , } |c_{n,i}^{(2)}| \leq \frac{\Delta}{2} c_{i,n}^{(1)} \ \forall n, i \quad (3)$$

Here we denote by $\mathbf{F}$ the matrix with columns $\{f_{n,j}(\mathbf{t})\}$ and $\mathbf{F}'$ the matrix with columns $\{f_{n,j}'(\mathbf{t})\}$. The $\ell_1$ penalty encourages sparsity, pushing most of the estimated amplitudes to zero, with higher $\lambda$ encouraging greater sparsity. Then, for each $(n,j)$ such that $c_{n,j}^{(1)} \neq 0$, one estimates that there is a waveform in the shape of $f_n$ with amplitude $\hat{a} = c_{n,j}^{(1)}$ and time shift $j\Delta - \hat{\tau} = j\Delta - c_{n,j}^{(2)}/c_{n,j}^{(1)}$ present in the signal. The inequality constraints in the optimization problem ensure first that we only recover positive amplitudes $\hat{a}$, and second that estimates $\hat{\tau}$ satisfy $|\hat{\tau}| < \Delta/2$. Requiring $\hat{\tau}$ to fall in this range keeps the estimated $\tau$ in the time bin represented by $f_{n,j}$ and also in the regime where they Taylor approximation to $f_{n,j}(t-\tau)$ is accurate. Note that (3) is a convex optimization problem.

Better results in [4] are obtained for a second order Taylor interpolation and the best results come from a polar interpolator, which represents each manifold of time-shifted waveforms $f_{n,j}(t - \tau), |\tau| \leq \Delta/2$ as an arc of the circle that is uniquely defined to pass through $f_{n,j}(t), f_{n,j}(t - \Delta/2)$, and $f_{n,j}(t+\Delta/2)$. Letting the radius of the arc be $r$, and its angle be $2\theta$ one represents points on this arc by linear combinations of functions $w, u, v$: $f(t-\tau) \approx w(t) + r\cos(\frac{2\tau}{\Delta}\theta)u(t) + r\sin(\frac{2\tau}{\Delta}\theta)v(t)$.

The Taylor and polar bases consist of shifted copies of elements chosen in order to linearly interpolate the curve in function space defined by $f_n(t - \tau)$ as $\tau$ varies from $-\Delta/2$ to $\Delta/2$. Let $\mathbf{G}_{n,k}$ be the matrix whose columns are $g_{n,j,k}(\mathbf{t})$ for $j \in 1, ..., N_\Delta$. With choices of basis elements, interpolator, and corresponding convex constraint set $C$ in place, one proceeds to estimate coefficients in the chosen basis by solving:

$$\text{argmin}_{\mathbf{c}} \left\| \mathbf{y} - \sum_{n=1}^{N_f} \sum_{k=1}^{K} \mathbf{G}_{n,k}\mathbf{c}_n^{(k)} \right\|_2^2 + \lambda \| \sum_{n=1}^{N_f} \mathbf{c}_n^{(1)} \|_1 \text{ subject to } (\mathbf{c}_{n,j}^{(1)}, ..., \mathbf{c}_{n,j}^{(K)}) \in C \ \forall (n,j) \quad (4)$$

One then maps back from each nonzero K-tuple of recovered coefficients $c_{n,j}^{(1)}, ..., c_{n,j}^{(K)}$ to corresponding $\hat{a}_{n,j}, \hat{\tau}_{n,j}$ that represent the amplitude and timing of the $n$th waveform present in the $j$th time bin. This can be done by inverting $\Phi$, if possible, or estimating $(\hat{a}_{n,j}, \hat{\tau}_{n,j}) = \text{argmin}_{a,\tau} \|\Phi(a,\tau) - (c_{n,j}^{(1)}, ..., c_{n,j}^{(K)})\|_2^2$.

Table 1: Basis choices (see also [4], Table 1.)

| Interpolator | Basis Vectors | $\Phi(a, \tau)$ | C |
|---|---|---|---|
| Taylor (K=3) | $\{f_{n,j}(\mathbf{t})\}, \{f'_{n,j}(\mathbf{t})\}, \{f''_{n,j}(\mathbf{t})\}$ | $(a, -a\tau, a\frac{\tau^2}{2})$ | $c^{(1)}, c^{(3)} > 0, |c^{(2)}| < c^{(1)}\frac{\Delta}{2}, \\ |c^{(3)}| < c^{(1)}\frac{\Delta^2}{8}$ |
| Polar | $\{\mathbf{w}_{n,j}\}, \{\mathbf{u}_{n,j}\}, \{\mathbf{v}_{n,j}\}$ | $(a, ar\cos(\frac{2\tau}{\Delta}\theta), \\ ar\sin(\frac{2\tau}{\Delta}\theta))$ | $c^{(1)} \geq 0, \sqrt{(c^{(2)})^2 + (c^{(3)})^2} \leq rc^{(1)} \\ rc^{(1)}\cos(\theta) \leq c^{(2)} \leq rc^{(1)}$ |
| SVD | $\{\mathbf{u}^1_{n,j}\}...\{\mathbf{u}^K_{n,j}\}.$ | (See Section 3.1) | (See Section 3.1) |

# 3 Continuous Orthogonal Matching Pursuit

We now present our method for recovery, which makes use of the idea of augmented bases presented above, but differs from CBP in several important ways. First, we introduce a different choice of basis that we find enables more accurate estimates. Second, we make use of a greedy method that iterates between choosing basis vectors and estimating time shifts and amplitudes, rather than proceeding via a single convex optimization problem as CBP does. Lastly, we introduce an alternative to the step of mapping back from recovered coefficients via $\Phi$ that notably improves the accuracy of the recovered time estimates.

Greedy methods such as Orthogonal Matching Pursuit (OMP) [11], Subspace Pursuit [2], and Compressive Sampling Matching Pursuit (CoSaMP) [8] have proven to be fast and effective in the realm of compressed sensing. Since the number of iterations of these greedy methods tend to go as the sparsity (when the algorithms succeed), they tend to be extremely fast when for very sparse signals. Moreover, our the greedy method eliminates the need to choose a regularization constant $\lambda$, a choice that can vastly alter the effectiveness of CBP. (We still need to choose $K$ and $\Delta$.) Our method is most closely analogous to OMP, but recovers continuous time estimates, so we call it Continuous Orthogonal Matching Pursuit (COMP). However, the steps below could be adapted in a straightforward way to create analogs of other greedy methods.

## 3.1 Choice of finite basis

We build upon [4], choosing as our basis $N_\Delta$ shifted copies of a set of $K$ basis vectors for each waveform in such away that these $K$ basis vectors can effectively linearly interpolate $f_n(t - \tau)$ for $|\tau| < \Delta/2$. In our method, as in Continuous Basis Pursuit, these basis vectors allow us to represent continuous time shifts instead of discrete time shifts, and expand the descriptive power of our dictionary without introducing undue amounts of coherence. While previous work introduced Taylor and polar bases, we obtain the best recovery from a different basis, which we describe now.

The basis comes from a singular value decomposition of a matrix whose columns correspond to discrete points on the curve in function space traced out by $f_{n,j}(t - \tau)$ as we vary $\tau$ for $|\tau| < \Delta/2$. Within one time bin of size $\Delta$, consider discretizing further into $N_\delta = \Delta/\delta$ time bins of size $\delta \ll \Delta$. Let $\mathbf{F}_\delta$ be the matrix with columns that are these (slightly) shifted copies of the waveform, so that the $i^{th}$ column of $\mathbf{F}_\delta$ is $f_{n,j}(\mathbf{t} - i\delta + \Delta/2)$ for a discrete vector of time points $\mathbf{t}$. Each column of this matrix is a discrete point on the curve traced out by $f_{n,j}(\mathbf{t} - \tau)$ as $\tau$ varies.

In choosing a basis, we seek the best choice of $K$ vectors to use to linearly interpolate this curve. We might instead seek to solve the related problem of finding the best $K$ vectors to represent these finely spaced points on the curve, in which case a clear choice for these $K$ vectors is the first $K$ left singular vectors of $\mathbf{F}_\delta$. This choice is optimal in the sense that the singular value decomposition yields the best rank-$K$ approximation to a matrix. If $\mathbf{F}_\delta = \mathbf{U}\mathbf{\Sigma}\mathbf{V}^\mathbf{T}$ is the singular value decomposition, and $\mathbf{u^k}, \mathbf{v^k}$ are the columns of $\mathbf{U}$ and $\mathbf{V}$ respectively, then $\|\mathbf{F}_\delta - \sum_{k=1}^K \mathbf{u}^k \Sigma_{k,k} (\mathbf{v}^k)^T\| \leq \|\mathbf{F} - \mathbf{A}\|$ for any rank-$K$ matrix $\mathbf{A}$ and any unitarily invariant norm $\|\cdot\|$.

In order to use this SVD basis with CBP or COMP, one must specify a convex constraint set for the coefficients of this basis. Since $af_{n,j}(t - i\delta) = \sum_{k=1}^{K} a\mathbf{u}^k \Sigma_{k,k} v_i^k$ a reasonable and simply enforced constraint set would be to assume that the recovered coefficients $c^{(k)}$ corresponding to each basis vector $\mathbf{u}^k$, when divided by $c^{(1)}$ to account for scaling, be between $\min_i \Sigma_{k,k} v_i^k$ and $\max_i \Sigma_{k,k} v_i^k$. A simple way to recover $a$ and $\tau$ would to choose $\tau = i\delta$ and $a, i$ to minimize $\sum_{k=1}^{K} (c^{(k)} - a\Sigma_{k,k} v_i^k)^2$.

In figure 3.1, we compare the error between shifted copies of a sample waveform $f(t - \tau)$ for $|\tau| < 0.5$ and the best (least-squares) approximation of that waveform as a linear combination of $K = 3$ vectors from the Taylor, polar, and SVD bases. The structure of the error as a function of the time shift $\tau$ reflects the structure of these bases. The Taylor approximation is chosen to be exactly accurate at $\tau = 0$ while the polar basis is chosen to be precisely accurate at $\tau = 0, \Delta/2, -\Delta/2$. The SVD basis gives the lowest mean error across time shifts.

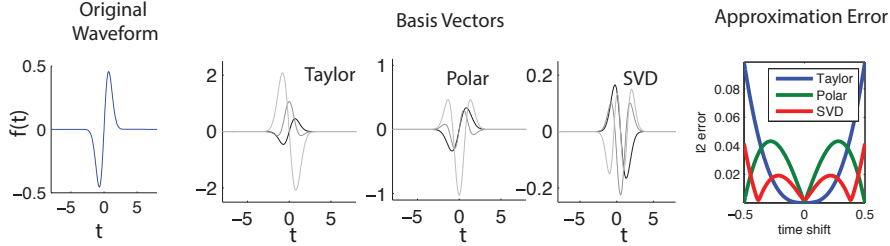

Figure 1: Using sample waveform $f(t) \propto t \exp(-t^2)$ (left panel), we compare the error introduced by approximating $f(t - \tau)$ for varying $\tau$ with a linear combination of $K = 3$ basis vectors, from the Taylor, polar or SVD bases. Basis vectors are shown in the middle three panels, and error in the far right panel. The SVD basis introduces the least error on average over the shift $\tau$. The average errors for the Taylor, polar, and SVD bases are 0.026, 0.027, and 0.014 respectively.

## 3.2 Greedy recovery

Having chosen our basis, we then greedily recover the time bins in which an occurrence of each waveform appears to be present. We would like to build up a set of pairs $(n, j)$ corresponding to an instance of the $n^{th}$ waveform in the $j^{th}$ time bin. (In our third step, we will refine the estimate within the chosen bins.)

Our greedy method is motivated by Orthogonal Matching Pursuit (OMP), which is used to recover a sparse solution $\mathbf{x}$ from measurements $\mathbf{y} = \mathbf{Ax}$. In OMP [11], one greedily adds a single dictionary element to an estimated support set $S$ at each iteration, and then projects orthogonally to adjust the coefficients of all chosen dictionary elements. After initializing with $S = \emptyset, \mathbf{x} = 0$, one iterates the following until a stopping criterion is met:

$$\mathbf{r} = \mathbf{y} - \mathbf{Ax}$$
$$j = \operatorname{argmax}_j \{ |\langle \mathbf{a}_j, \mathbf{r} \rangle| \; s.t. \; j \in \{1, ...J\} \backslash S \}$$
$$S = S \cup \{j\}$$
$$\mathbf{x} = \operatorname{argmin}_{\mathbf{z}} \{ ||\mathbf{y} - \mathbf{Az}||_2 \; s.t. \; z_i = 0 \; \forall \, i \notin S \}$$

If we knew the sparsity of the signal, we could use that as our stopping condition. Normally we do not know the sparsity a priori; we stop when changes in the residual become sufficiently small.

We adjust this method to choose at each step not a single additional element but rather a set of $K$ associated basis vectors. $S$ is again initialized to be empty, but at each step we add a time-bin/waveform pair $(n, j)$, which is associated with $K$ basis vectors. In this way, we are adding $K$ vectors at each step, instead of one as in OMP. We greedily add the next index $(n, j)$ according to:

$$(n, j) = \operatorname{argmin}_{m,i} \left\{ \min_{\mathbf{c}_{m,i}} \{ || \sum_{i=1}^{k} c_{m,i}^{(k)} \mathbf{g}_{m,i}^{(k)} - \mathbf{r} ||_2^2 \; s.t. \; \mathbf{c}_{m,i} \in C \} \, , \, (m, i) \in S^c \right\} \qquad (5)$$

Here $\{\mathbf{g}_{m,i}^{(k)}\}$ are the chosen basis vectors (Taylor, polar, or SVD), and $C$ is the corresponding constraint set, as in Section 2.

In comparison with the greedy step in OMP, choosing $j$ as in (5) is more costly, because we need to perform a constrained optimization over a $K$ dimensional space for each $n, j$. Fortunately, it is not necessary to repeat the optimization for each of the $N_f \cdot N_\Delta$ possible indices each time we add an index. Assuming waves are localized in time, we need only update the results of the constrained optimization locally. When we update the residual $\mathbf{r}$ by subtracting the newly identified waveform $n$ in the $j^{th}$ bin, the residual only changes in the bins at or near the $j^{th}$ bin, so we need only update the quantity $\min_{\mathbf{c}_{n,j'}} \{\| \sum_{i=1}^{k} c_{n,j'}^{(k)} \mathbf{g}_{n,j'}^{(k)} - \mathbf{r} \|_2^2 \, s.t. \, \mathbf{c}_{n,j'} \in C \}$ for $j'$ neighboring $j$.

### 3.3 Estimating time shifts

Having greedily added a new waveform/timebin index pair $(n, j)$, we next define our update step, which will correspond to the orthogonal projection in OMP. We present two alternatives, one of which most closely mirrors the corresponding step in OMP, the other of which works within the Fourier domain to obtain more accurate recovery.

To most closely follow the steps of OMP, at each iteration after updating $S$ we update coefficients $\mathbf{c}$ according to:

$$\mathrm{argmin}_c \left\| \sum_{(n,j) \in S} \sum_{k=1}^{K} c_{n,j}^{(k)} \mathbf{g}_{n,j}^{(k)} - \mathbf{y} \right\|_2^2 \text{ subject to } \mathbf{c}_{n,j} \in C \; \forall \, (n,j) \in S \tag{6}$$

We alternate between the greedily updating $S$ via (5), and updating $\mathbf{c}$ as in (6), at each iteration finding the new residual $\mathbf{r} = \sum_{(n,j) \in S} \sum_{k=1}^{K} c_{n,j}^{(k)} \mathbf{g}_{n,j}^{(k)} - \mathbf{y}$ ) until the $\ell_2$ stopping criterion is reached. Then, one maps back from $\{\mathbf{c}_{n,j}\}_{(n,j) \in S}$ to $\{a_{(n,j)}, \tau_{(n,j)}\}_{(n,j) \in S}$ as described in Section 2.

Alternatively we may replace the orthogonal projection step with a more accurate recovery of spike timings that involves working in the Fourier domain. We use the property of the Fourier transform with respect to translation that: $(f(t - \tau))^\wedge = e^{2\pi i \tau} \hat{f}$. This allows us to estimate $a, \tau$ directly via:

$$\mathrm{argmin}_{a,\tau} \|( \sum_{n,j \in S} a_{n,j} e^{2\pi i \boldsymbol{\omega} \tau_{n,j}} \hat{f}_{n,j}(\boldsymbol{\omega})) - \hat{y}(\boldsymbol{\omega}) \|_2 \text{ subject to } |\tau_{n,j}| < \Delta/2 \; \forall \, (n,j) \in S \tag{7}$$

This is a nonlinear and non-convex constrained optimization problem. However, it can be solved reasonably quickly using, for example, trust region methods. The search space is dramatically reduced because $\tau$ has only $|S|$ entries, each constrained to be small in absolute value. By searching directly for $a, \tau$ as in (7) we sacrifice convexity, but with the benefit of eliminating from this step error of interpolation introduced as we map back from $\mathbf{c}$ to $a, \tau$ using $\Phi^{-1}$ or a least squares estimation.

It is easy and often helpful to add inequality constraints to $a$ as well, for example requiring $a$ to be in some interval around 1, and we do impose this in our spike-sorting simulations and analysis in Section 4. Such a requirement effectively imposes a uniform prior on $a$ over the chosen interval. It would be an interesting future project to explore imposing other priors on $a$.

## 4 Results

We test COMP and CBP for each choice of basis on simulated and neural data. Here, COMP denotes the greedy method that includes direct estimation of $a$ and $\tau$ during the update set as in (7). The convex optimization for CBP is implemented using the cvx package for MATLAB [7], [6].

### 4.1 Simulated data

We simulate a signal $y$ as the sum of time-shifted copies of two sample waveforms $f_1(t) \propto t \exp(-t^2)$ and $f_2(t) \propto e^{-t^4/16} - e^{-t^2}$ (Figure 2a). There are $s_1 = s_2 = 5$ shifted copies of $f_1$ and $f_2$, respectively. The time shifts are independently generated for each of the two waveforms using a Poisson process (truncated after 5 spikes), and independent Gaussian noise of variance $\sigma^2$ is

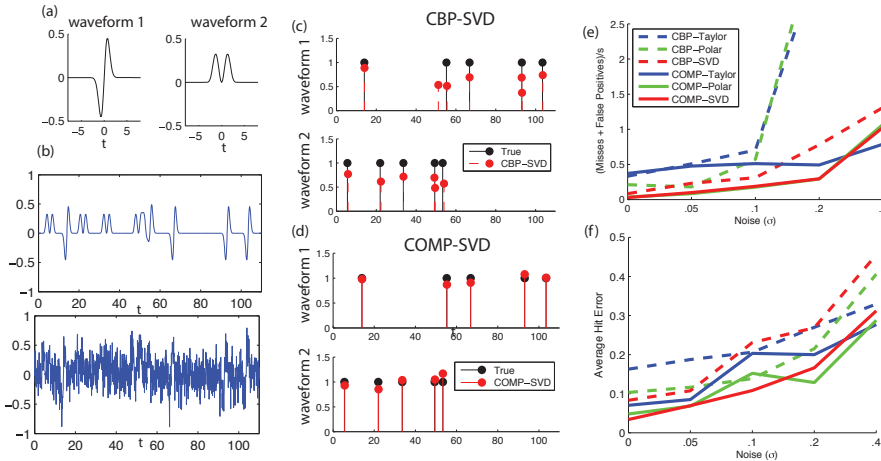

Figure 2: (a) Waveforms present in the signal. (b) A noiseless (top) and noisy (bottom) signal with $\sigma = .2$. (c) Recovery using CBP. (d) Recovery using COMP (with $a$, $\tau$ updated as in (7)). (e) For each recovery method over different values of the standard deviation of the noise $\sigma$, misses plus false positives, divided by the total number of events present, $s = s_1 + s_2$. (f) Average distance between the true and estimated spike for each hit.

added at each time point. Figures 2b,c show an example noise-free signal ($\sigma = 0$), and noisy signal ($\sigma = .2$) on which each recovery method will be run.

We run CBP with the Taylor and polar bases, but also with our SVD basis, and COMP with all three bases. Since COMP here imposes a lower bound on $a$, we also impose a thresholding step after recovery with CBP, discarding any recovered waveforms with amplitude less than .3. We find the thresholding generally improved the performance of the CBP algorithm by pruning false positives. Throughout, we use $K = 3$, since the polar basis requires 3 basis vectors per bin.

We categorize hits, false positive and misses based on whether a time shift estimate is within a threshold of $\epsilon = 1$ of the true value. The "average hit error" of Figure 2h, 3b is the average distance between the true and estimated event time for each estimate that is categorized as a hit. Results are averaged over 20 trials.

We compare CBP and COMP over different parameter regimes, varying the noise ($\sigma$) and the bin size ($\Delta$). Figures 2g and 3a show misses plus false positives for each method, normalized by the total number of events present. Figures 2f and 3b show average distance between the true and estimated spike for each estimate categorized as a hit. The best performance by both measures across nearly all parameter regimes considered is achieved by COMP using the SVD basis. COMP is more robust to noise (Figure 2g), and also to increases in bin width $\Delta$. Since both algorithms are faster for higher $\Delta$, robustness with respect to $\Delta$ is an advantage. We also note a significant increase in CBP's robustness to noise when we implement it with our SVD basis rather than with the Taylor or polar basis (Figure 2e).

A significant advantage of COMP over CBP is its speed. In Figure 3c we compare the speed of COMP (solid) and CBP (dashed) algorithms for each basis. COMP yields vast gains in speed. The comparison is especially dramatic for small $\Delta$, where results are most accurate across methods.

## 4.2   Neural data

We now present recovery of spike times and identities from neural data. Recordings were made using glass-coated tungsten electrodes in the lateral intraparietal sulcus (LIP) of a macaque monkey performing a motion discrimination task. In addition to demonstrating the applicability of COMP to sorting spikes in neural data, this section also shows the resistance of COMP to a certain kind of error that recovery via CBP can systematically commit, and which is relevant to neural data.

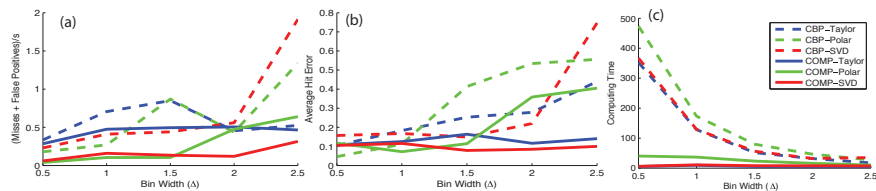

Figure 3: (a) Misses plus false positives, divided by the total number of events present, $s = s_1 + s_2$ over different values of bin width $\Delta$. (b) Average distance between the true and estimated spike for each hit for each recovery method. (c) Run time for COMP (solid) and CBP (dashed) for each basis.

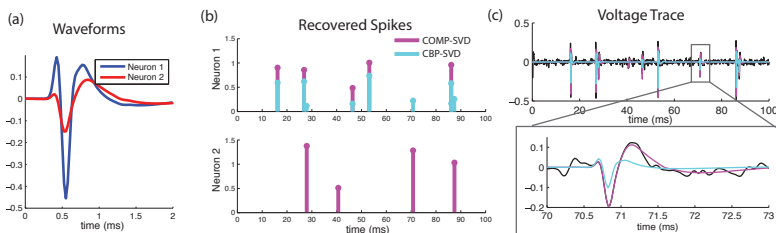

Figure 4: (a) Two neural waveforms; each is close to as scaled copy of the other (b) Recovery of spikes via COMP (magenta) and CBP (cyan) using the SVD basis. CBP tends to recover small-amplitude instances of waveform one where COMP recovers large amplitude instances of waveform two (c) Top: recovered traces. Lower panel: zooming in on an area of disagreement between COMP and CBP. The large-ampltude copy of waveform two more closely matches the trace

In the data, the waveform of one neuron resembles a scaled copy of another (Figure 4a).The similarity causes problems for CBP or any other $\ell_1$ minimization based method that penalizes large amplitudes. When the second waveform is present with an amplitude of one, CBP is likely to incorrectly add a low-amplitude copy of the first waveform (to reduce the amplitude penalty), instead of correctly choosing the larger copy of the second waveform; the amplitude penalty for choosing the correct waveform can outweigh the higher $\ell_2$ error caused by including the incorrect waveform.

This misassignment is exactly what we observe (Figure 4b). We see that CBP tends to report small-amplitude copies of waveform one where COMP reports large-amplitude copies of waveform two. Although we lack ground truth, the closer match of the recovered signal to data (Figure 4c) indicates that the waveform identities and amplitudes identified via COMP better explain the observed signal.

## 5  Discussion

We have presented a new greedy method called Continuous Orthogonal Matching Pursuit (COMP) for identifying the timings and amplitudes for waveforms from a signal that has the form of a (noisy) sum of shifted and scaled copies of several known waveforms. We draw upon the method of Continuous Basis Pursuit, and extend it in several ways. We leverage the success of Orthogonal Matching Pursuit in the realm of sparse recovery, use a different basis derived from a singular value decomposition, and also introduce a move to the Fourier domain to fine-tune the recovered time shifts. Our SVD basis can also be used with CBP and in our simulations it increased performance of CBP as compared to previously used bases. In our simulations COMP obtains increased accuracy as well as greatly increased speed over CBP across nearly all regimes tested. Our results suggest that greedy methods of the type introduced here may be quite promising for, among other applications, spike-sorting during the processing of neural data.

### Acknowledgments

This work was supported by the McKnight Foundation (JP), NSF CAREER Award IIS-1150186 (JP), and grants from the NIH (NEI grant EY017366 and NIMH grant MH099611 to AH & JP).

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
