[Reviews · NeurIPS 2014]

Submitted by Assigned_Reviewer_6

This paper presented a new method for spike-sorting, although the method itself is general and not limited to spike-sorting, and was presented in a generic way.

The authors introduced two new ideas. First is to use SVD to define basis functions and compute precise spike times by interpolation, which is I think a very neat idea. To this end, Taylor or polar interpolators are previously used (Ekanadham et al., 2011), but the authors pointed out that SVD-based method is theoretically optimal in MSE sense.

Second is to employ a greedy algorithm instead of a convex optimization solver. This is similar to OMP, but the authors further proposed an alternative tweak for finding finer spike times by working in the Fourier domain. As a result, computing time is greatly improved (Figure 3c).

I think this is a very nice paper overall.

Minor comments:
- There are several erroneous stentences (e.g., line 51; maybe 195-196; 429).
- What is the unit of noise in Figure 2(g) and (h)?
- It would be much nicer if more thorough examinations of the proposed model with spike-sorting problems were presented, either with simulated or actua recordingl data.
Summary: This paper presented a new spike-sorting algorithm that improved both performance and computational efficiency.

Submitted by Assigned_Reviewer_38

This paper proposes a new algorithm of signal recovery which combines the advantages of continuous basis pursuit (CBP) and orthogonal matching pursuit (OMP).
This algorithm chooses basis vectors from left singular vectors from SVD of matrix of dictionaries, in order to better interpolate the data point between the grid. And simulation results show the SVD basis outperforms previous Taylor and Polar basis. Then, motivated by orthogonal matching pursuit, the algorithm greedily recovers the time bins of occurrence of each waveform, and moves to the Fourier domain to more precisely estimate time shifts.

The paper is clearly written and seems new in this field. And the new algorithm outperforms previous ones.
Summary: A new algorithm about signal recovery.

Submitted by Assigned_Reviewer_44

"Inferring sparse representations of continuous signals with continuous orthogonal matching pursuit" is a nice, compact, well-written paper on a method that should be of relevance for anyone dealing with spike sorting. The authors build up on work of Ekanadham and all (2011a,b) by considering an alternative basis for decompositions, and a different optimisation method. They provide comparison with previous results and report significant improvement in computation time, and evidence for increased accuracy of detection and estimation of waveform parameters.

One question about the comparison - in order to be fair to the older method, the authors pruned waveforms with amplitude <.3. Was that value optimized for (was it equal to the lower bound on a in COMP case)? Could CBP score better, if that parameter was further increased? How was lambda of the L1-penalty chosen?
Also - is it possible to plot hits and misses in separate plots? Or are these not telling more about the nature of typical errors of various methods?

Are all "hit errors" of COMP calculated from the "Fourier" method?

Figure 1
Might be a bit confusing to the reader - having observed the original waveform, I expected to see the reconstructed waveform. It took me a while to realise the black shapes in panels 2-4 were not the reconstructions. I would suggest to at least expand the legend.
The average errors provided in the legend seem to be much lower than values plotted in the figure.

Since the paper is so well written, I include suggestion to double-check lines:
41 (spurious "the")
85 (spurious "here")
153 (theta missing from the argument of sinus)
299/300 please, correct all indices (j' and n are swapped, (i,j) came out of nowhere)
311 and 315 - indices swapped
411 eps=1 - do you mean 1 Delta?
Summary: A nice, compact, well-written paper on a method that should be of relevance for anyone dealing with spike sorting.
Author Feedback
Author rebuttal: We thank all the reviewers for their careful reading of the paper and valuable feedback. Detailed replies to each comment are below.

Assigned Reviewer_44
------------------

> authors pruned waveforms with amplitude <.3. Was >that value optimized for (was it equal to the lower >bound on a in COMP case)? Could CBP score better, if >that parameter was further increased?

We discounted recovered amplitudes from CBP that were lower than .3, with the bound chosen to match the bound on the amplitude (a) enforced by our implementation of COMP. The fact that recovered waveforms of small amplitudes must be pruned for optimal performance has been previously observed in [Ekanadham, C. (2012). Continuous basis pursuit and its applications (Doctoral dissertation, New York University). ], where the author used a threshold of .5. We set the threshold to .3 based on an initial run in which it appeared to perform better than higher values, but we will check more rigorously in the revision.

> How was lambda of the L1-penalty chosen?

We chose lambda through cross-validation, testing along a range of values of lambda and choosing the one that minimized hits + false positives.

> is it possible to plot hits and misses in separate >plots? Or are these not telling more about the nature >of typical errors of various methods?

We originally plotted hits and misses in separate plots, but subsequently combined them both for reasons of space and because the separate plots seemed to convey little additional information. We didn't observe any interesting differences in the types of errors between the methods.

>Are all "hit errors" of COMP calculated from the >"Fourier" method?

Yes, all of the hit errors of COMP were calculated from the “Fourier” method. We will add this to the legend or caption in the final version of the paper for more clarity.

> Figure 1: Might be a bit confusing to the reader - >having observed the original waveform, I expected to >see the reconstructed waveform. It took me a while to >realise the black shapes in panels 2-4 were not the >reconstructions. I would suggest to at least expand >the legend.

In figure 1 we can certainly make it clearer for the reader that panels 2-4 show the basis vectors by expanding the legend and caption.

>The average errors provided in the legend seem to be >much lower than values plotted in the figure.

Thank you- this is a mistake that we will fix in the final paper. The average errors in the legend were from a different version of the plot that was subsequently replaced.

>suggestion to double-check lines:

We appreciate the careful reading and suggestion to double-check these lines.

Assigned_Reviewer_6
------------------

> What is the unit of noise in Figure 2(g) and (h)?

Noise in Figure 2g and 2h is measured by sigma, the standard deviation of the independent mean-zero Gaussian noise that was added to each time step of the noise-free signal. We note this in lines 375-377, and
will add it to the figure caption in the final version of the paper to make it clearer to the reader.

> It would be much nicer if more thorough examinations >of the proposed model with spike-sorting problems >were presented, either with simulated or actual >recording data.

We appreciate the feedback that a more thorough analysis of the method in spike-sorting tasks would improve the paper. Analysis of the performance of the method on actual neural data was not complete at
the time of submission, but we have since performed such analysis and intend to include it in the final paper.

>Suggestion to check specific lines for errors

We appreciate the careful reading and suggestion to double-check specific lines.